# L1 Cell Adhesion Molecule in Cancer, a Systematic Review on Domain-Specific Functions

**DOI:** 10.3390/ijms20174180

**Published:** 2019-08-26

**Authors:** Miriam van der Maten, Casper Reijnen, Johanna M.A. Pijnenborg, Mirjam M. Zegers

**Affiliations:** 1Department of Obstetrics and Gynaecology, Radboud university medical center, 6525 GA Nijmegen, The Netherlands; 2Department of Cell Biology, Radboud Institute for Molecular Life Sciences, Radboud university medical center, 6525 GA Nijmegen, The Netherlands; 3Department of Obstetrics and Gynaecology, Canisius-Wilhelmina Hospital, 6532 SZ Nijmegen, The Netherlands

**Keywords:** L1CAM, tumor biology, cell adhesion, systematic review, oncogenic signaling, biomarker

## Abstract

L1 cell adhesion molecule (L1CAM) is a glycoprotein involved in cancer development and is associated with metastases and poor prognosis. Cellular processing of L1CAM results in expression of either full-length or cleaved forms of the protein. The different forms of L1CAM may localize at the plasma membrane as a transmembrane protein, or in the intra- or extracellular environment as cleaved or exosomal forms. Here, we systematically analyze available literature that directly relates to L1CAM domains and associated signaling pathways in cancer. Specifically, we chart its domain-specific functions in relation to cancer progression, and outline pre-clinical assays used to assess L1CAM. It is found that full-length L1CAM has both intracellular and extracellular targets, including interactions with integrins, and linkage with ezrin. Cellular processing leading to proteolytic cleavage and/or exosome formation results in extracellular soluble forms of L1CAM that may act through similar mechanisms as compared to full-length L1CAM, such as integrin-dependent signals, but also through distinct mechanisms. We provide an algorithm to guide a step-wise analysis on L1CAM in clinical samples, to promote interpretation of domain-specific expression. This systematic review infers that L1CAM has an important role in cancer progression that can be attributed to domain-specific forms. Most studies focus on the full-length plasma membrane L1CAM, yet knowledge on the domain-specific forms is a prerequisite for selective targeting treatment.

## 1. Introduction

L1 cell adhesion molecule (L1CAM) is a 200–220 kDa transmembrane glycoprotein. In its mature, full length form (L1CAM-FL; FL), it has a long ectodomain that comprises six immunoglobulin-like (Ig) domains followed by five fibronectin type III repeats. It further contains a single transmembrane domain and a relatively short highly conserved cytoplasmic domain (Figure 1) [1,2]. L1CAM is mostly known for its role in neural development, by its ability to regulate processes such as neurite outgrowth, fasciculation, cell adhesion, cell migration, myelination and cell survival [3]. These diverse effects depend on homophilic interaction with other L1CAM molecules [4,5] or heterophilic interactions of the L1CAM ectodomain in both *cis* and *trans* with other neural cell adhesion molecules such as integrins or CD24, and other binding partners such as neurocan or neuropilin-1 [6,7,8] (Figure 1). In addition, the cytosolic domain of L1CAM can interact with several different binding partners, including AP2, CKII, P90rsk, FAK and ezrin, which controls its expression on the membrane through endocytosis, mediates interaction with the cytoskeleton, and activates downstream signaling pathways [2]. The diverse roles of L1CAM further depend on its different cellular expression forms. Differential expression can be due to variant isoform expression by alternative splicing of exon 2 [9] or exon 27 [10], which affects homophilic binding and endocytosis, respectively, whereas exclusion of exon 25, gives rise to a soluble isoform that lacks the entire transmembrane domain [11].

In addition to pathways that act downstream of membrane-bound L1CAM-FL, many L1CAM-mediated processes can be induced by proteolytic cleavage-products of L1CAM-FL, (Figure 2A,B). L1CAM cleavage is mediated by the metalloproteases a disintegrin and metalloproteinase domain-containing protein 10 (ADAM10) [12], and ADAM17 [13], or matrix metalloprotease 16 (MMP-16) [14] which produce a soluble ectodomain of ~200 kDa that is shed in the extracellular environment (L1CAM-ECD), and a membrane-bound cytosolic domain with an apparent molecular weight of 32 kDa that is further processed by intracellular presenilin to a 28 kDa soluble cytoplasmic domain (LICAM-CT) that can transfer to the nucleus [13,15]. Besides extracellular L1CAM as generated by proteolytic cleavage [12,13], or potentially by alternative splicing [11], another pool of extracellular L1CAM could be derived from exosomes or microvesicles [16,17]. (Figure 1). Both exosomes, which are derived from exocytosed multivesicular endosomes, and microvesicles, which bud directly from the plasma membrane, maintain an exoplasmic membrane topology [18], and exosomal L1CAM thus will still expose its ectodomain towards the extracellular environment, similar to cell-associated L1CAM.

In addition to its role in neurogenesis, L1CAM is involved in tumor progression of multiple cancers. High L1CAM expression is associated with advanced tumor stages, metastases and poor prognoses [19,20,21]. L1CAM was demonstrated to be involved in different pro-tumor events such as metastasis, epithelial-to-mesenchymal transition (EMT), and is associated with aggressive tumor phenotypes, and chemoresistance [19,22,23,24,25,26]. Besides the prognostic value of L1CAM, the protein is considered to be suitable for targeted therapy because of its role in tumor progression [27,28,29,30]. Indeed, function-blocking antibodies, targeting the ECD have shown to inhibit tumor cell growth in vivo [31] and in mouse models [28,32,33]. Similar as in neurons, tumor-associated L1CAM is proteolytically cleaved and soluble ectodomains can be detected in the sera and ascites of ovarian [16,19,34] and in the sera of gastrointestinal [35] and breast [36] cancer patients. Proteolytic cleavage is affected by growth factors such as HGF [37] and is regulated by ERK- and Src-controlled mechanism [38] and further depends on the level of glycosylation of its ectodomain [22,39] and on interactions of the cytoplasmic domain with ankyrin and the cytoskeleton [37]. Finally, exosomal L1CAM is produced by patient-derived primary GBM cells [40,41] and can be recovered from blood and ascites from ovarian cancer patients [42] and may promote as platform L1CAM cleavage or systemic delivery vehicle of membrane-bound L1CAM forms [43].

Both L1CAM-FL and its cleavage products exert biological functions, but the specific contributions of differentially processed L1CAM forms in the plasma membrane, the intracellular or extracellular environment to cancer progression are incompletely understood. Since L1CAM is a clinically valuable prognostic marker it is essential to understand what form is analyzed in standard and clinical used assays. Moreover, as L1CAM is a potential target for novel cancer therapies, understanding specific domain functions is important for targeted therapy. The aims of this review are to determine the domain-specific functions of L1CAM in cancer progression and to identify the gaps in knowledge relevant for clinical implications. For this, we have performed a systematic review on current literature, in which we have specifically screened for publications in which L1CAM functions were linked to information on relevant protein domain and on linked downstream signaling pathways in tumor cells. We further provide an algorithm that facilitates the translation of the available knowledge of domain-specific L1CAM into clinical practice.

## 2. Results

In Table 1, all studies are categorized according to the studied form of L1CAM. Details of each study are listed in Appendix A.

### 2.1. Full-length L1CAM or General Functions of Cell-associated L1CAM

In a total of 19 studies investigated the impact of full-length L1CAM (L1CAM-FL) on tumor suppression with strategies that did not specifically address domain-specific functions of L1CAM [34,44,45,46,47,48,49,50,51,52,53,54,55,56,57,58,59,60,61]. A complete overview of the main study strategies and findings is outlined in Appendix A and a summary of results can be found in Figure 3. In general, the methodology of these studies involved screening of expression of patient samples and tumor cell lines across a wide array of mostly carcinomas, flanked by overexpression of L1CAM-FL or siRNA-mediated knockdown (KD) in vitro and in vivo. A majority of these studies show that L1CAM-FL promotes cell migration and invasion (*n* = 15). Additional tumor-promoting processes include stimulation of proliferation in vitro (*n* = 9) and tumor growth and metastasis in vivo. Additional tumor promoting processes include regulation of gene transcription and the increased radio- or chemoresistance and regulation of apoptosis. Finally, several studies indicate a role of L1CAM in cell-cell adhesion.

This systematic review selected for studies in which the mechanisms of L1CAM in tumor progression were addressed, and if possible, could be linked to a specific domain of L1CAM. The studies in which domain-specific functions were not specifically studied, demonstrate that tumor cell proliferation and tumor growth, and migration and invasion is often associated with activation of ERK, such as in esophageal squamous cell cancer (ESCC) [45], gastric cancer [47], non-small cell lung cancer (NSCLC) [54], ovarian carcinoma [55], melanoma [51] and pancreatic cancer (PC) [60]. Besides promoting proliferation, ERK-dependent gene expression downstream of L1CAM is reported for ESCC [45] and melanoma [51], resulting in gene expression involved in migration and invasion. In contrast to these publications, several other studies that investigated the role of ERK were unable to demonstrate its activation, but reported activation of JNK, p38 MAPK and/or Akt/PI3K, in respectively, gastic cancer [49], extrahepatic cholangiacarcinoma [46], retinoblastoma [56] or gall bladder cancer [48]. Activation of these pathways may also occur in parallel with ERK activation, for instance in ovarian carcinoma [55]. L1CAM is also reported to be associated with the Wnt/b-catenin pathway, both as target in colorectal carcinoma [26] but also as upstream regulator in melanoma [52] and breast cancer [44].

Radio- or chemoresistance is associated by treatment-induced upregulation of L1CAM in neuroblastoma [53] and pancreatic cancer [58,59]. Pathways involved in regulating treatment resistance and apoptosis as regulated by L1CAM include PI3K/Akt signaling in gastric cancer [49], retinoblastoma [56] and neuroblastoma [53]. Another level of regulation includes the control of caspases, as L1CAM suppresses transcription of caspase-8 in glioblastoma [50]. Furthermore, in pancreatic cancer, L1CAM promotes, via α5-integrin para- or autocrine signaling of IL1β, which induces iNOS activation and NOS secretion, leading to the inhibition of caspase 3 and 7 activity [58,59]. Since L1CAM was also reported to induce IL1β in ovarian cancer [34], L1CAM-induced resistance in this cancer may be regulated through a similar mechanism.

#### 2.1.1. Alternatively Spliced Variants

Three publications address the function of two isoforms of L1CAM that are generated due to alternative splicing of the L1CAM mRNA. It is reported that L1CAM-SV, which lacks exon 2 and 27, is co-expressed with L1CAM-FL, with L1CAM-SV being the dominant form in many tumor types [44,62]. Both L1CAM-SV and L1CAM-FL can promote motility in breast cancer cells [44]. Surprisingly however, L1CAM-FL, but not L1CAM-SV, confers metastasis in ovarian cancer, CRC, fibrosarcoma, and T-cell lymphoma [62]. A second, newly identified soluble isoform lacking the exon 25-encoded transmembrane domain, was recently identified in endothelial cells. This isoform is highly expressed in ovarian-cancer associated vessels and has an increased potential to induce angiogenesis, which relies on FGFR1 signaling [11].

#### 2.1.2. L1CAM-FL Studies With FL-ECD-specific Functions

Twelve studies addressed the function of the ECD of L1CAM-FL (FL-ECD), often with approaches that involved targeted mutations within this domain, flanked by KD experiments. [11,16,23,59,62,63,64,65,66,67,68,69]. Often, these studies also investigated the role of the soluble ECD after shedding, which will be discussed in Section 2.2.1, The ectodomain of L1CAM-FL engages in homotypic interactions [63] or heterotypically interacts with different binding partners such as integrins αvβ3, αvβ5 and α5β1, activated leukocyte cell adhesion molecule (ALCAM), E-selectin, neuropilin 1 and FGFR [11] in *cis* or *trans* to confer transendothelial migration, proliferation and invasion of cancer cells [23,51,63,64,66,67,68]. Among these interactions, L1CAM-integrin interactions, which depend on the RGD sequence in the sixth immunoglobulin-like domain of L1CAM, have been most extensively studied, both for their role in cell-cell interactions and/or intracellular signaling [23,51,59,63,64,66,67,68].

Melanoma cells may use L1CAM to bind αvβ3 on endothelial cells to promote transendothelial migration [66]. Conversely, breast cancer cells were shown to use α5β1 integrin to engage with upregulated L1CAM in tumor-associated fibroblasts [64], but do not rely on this integrin to interact with endothelia [63]. Alternative mechanism to engage with other cell types to mediate adhesion and transendothelial migration include ALCAM, which allows breast cancer cells to interact with endothelium [63], and NRP-1, which is involved in the interaction of ovarian cancer cells with the mesothelium [67].

The interaction between L1CAM-FL-ECD and integrins leads to activation of integrins and downstream phosphorylation and activation of focal adhesion kinase (FAK), proto-oncogene tyrosine-protein kinase Src, and integrin-linked kinase in pancreatic cancer [23,68]. In breast cancer integrin binding activates ERK [64], but in other tumor cells integrin-binding is dispensable for ERK activation [69]. Further downstream, integrins can activate nuclear factor kappa-light-chain-enhancer of activated B cells (NF-kB) in pancreatic cancer [23,68], but not in ovarian cancer [34]. Activation of NF-κB in colorectal cancer is negatively influenced by mutation of the H210 amino acid residue in the 2^nd^ Ig-like domain in the ECD [65], which suggest a role for homophilic interaction in NF-κB activation. Interestingly, mutation of H210 in the ECD abolishes the interaction of L1CAM cytoplasmic domain with ezrin [65]. The activation of NF-κB by L1CAM-FL through ezrin can activate the metalloendopeptidase CD10/neprilysin which subsequently confers proliferation, motility and metastatic capacity in colorectal cancer [65]. Since ezrin is an important mediator of L1CAM-dependent processes (see below) and is transcriptionally upregulated downstream of β1 integrins in CRC [45], it appears that one of the functions of the ECD is to transcriptionally regulate the cytoplasmic factors required for signal transduction towards tumor promoting processes, including proliferation, migration and metastasis.

#### 2.1.3. L1CAM-FL Studies with CT-specific Functions

Nine studies on L1CAM-FL focused on interactions and downstream functions of the cytoplasmic domain of the full length protein (FL-CT), mostly focusing on colorectal cancer, ovarian carcinoma and pancreatic cancer [23,26,27,62,68,69,70,71,72].

The cytoplasmic domain appears to be mainly involved in regulating NF-κB and ERK activity. The publications included in this review indicate that these events are independently regulated. Several studies report that the activation of NF-κB depends on the cytoskeletal linker protein ezrin. Ezrin binds directly to the L1CAM C-terminus, and its binding site has been mapped to involve a critical Y1151 residue [70]. The interaction of L1CAM with ezrin, and downstream activation of NF-κB is elevated in invasive colorectal tumor fronts [70] and is necessary for proliferation, invasion and metastasis of colorectal cancer cells [26,70,71]. A similar mechanism was reported in pancreatic and breast cancer, where the upstream signal involved in ezrin-mediated NF-κB activation involved β1-integrin [23,68].

As already indicated in Section 2.1, many L1CAM-dependent effects involve activation of ERK and forced clustering of cytoplasmic domains, using chimeric proteins, is sufficient to induce ERK activation [72] which suggests that ERK can be directly and autonomously activated by the L1CAM cytoplasmic domain. The C-terminal amino acid residues T1247, S1248 and N1251 are involved in ERK activation in pancreatic or ovarian cancer [27,69,73], by mechanisms that are not yet fully elucidated, but do not depend on binding to integrins [69] or to RanBPM [72], a protein involved in MAPK activation in non-tumor cells [73].

### 2.2. Extracellular L1CAM

#### 2.2.1. Soluble Ectodomain of L1CAM (L1CAM-ECD)

Twelve studies investigated the effects of the cleaved, soluble N-terminal ectodomain of L1CAM (L1CAM-ECD) on tumor progression, mostly focusing on gliomas, ovarian cancer, and pancreatic cancer [16,33,34,40,41,44,67,74,75,76,77,78]. Increased levels of L1CAM-ECD are caused by an upregulation of ADAM 10 in glioma, ovarian cancer and colon and cancer cells [16,26,40,41], and shedding is promoted by the presence of serum or HGF [33], hypoxia or apoptotic stimuli [16]. With the exception of breast cancer [44], L1CAM-ECD stimulated migration and invasion in all studies. In glioma, L1CAM-ECD promotes cell migration and proliferation by a mechanism that depends on αvβ3 and αvβ5 integrins and FAK, and on FGFR1 activity [40,41,74,75], suggesting that the L1CAM-ECD mediates its effect by heterophilic interactions of both integrins and the FGFR1 in these tumors. In ovarian cancer, L1CAM-ECD interacts with to α5β1, αvβ5, and αvβ3 integrins [16,77], and with NRP-1, the latter of which allows for binding the mesothelium [67]. L1CAM-ECD stimulated the activation of ERK, FAK and Src and provided resistance against apoptosis [16,33,76] and increased L1CAM-ECD in ovarian cancer patients correlates with increased chemoresistance [34]. Furthermore, heterotypic interactions of L1CAM-ECD from ovarian cancers and α3 integrins of endothelial cells, promotes angiogenesis through the vascular endothelial growth factor receptor 2 (VEGFR-2) [77]. Finally, a study focusing on pancreatic cancer demonstrated that, L1CAM-ECD cleaved from Schwann cells homotypically interacts with L1CAM on tumor cells leading to the activation of ERK and STAT3-mediated expression of MMP-2 and -9 which facilitates cancerous nerve invasion [78].

#### 2.2.2. Exosomal L1CAM

Three studies focused on the impact of exosomal L1CAM forms in glioma and ovarian cancer which potentially include LICAM-FL associated with exosomes (Exo-FL) or L1-CAM lacking either the cytoplasmic or the N-terminal domain (Exo-ΔCT and Exo-ΔNT, respectively) on tumor progression (Figure 3C). Two studies demonstrated that exosomes may contain both full-length (Exo-FL) or proteolyzed fragments (Exo-ΔCT/Exo-ΔECD) of L1CAM [16,40]. The latter are formed and released through similar mechanisms as the formation of the soluble ectodomain of L1CAM, by ADAM10 mediated proteolytic cleavage [16]. In addition to exosomal L1CAM, L1CAM can also be present in apoptotic membrane vesicles which are formed by multiple metalloproteinases in reaction to apoptotic stimuli or hypoxia [16]. In both glioma and ovarian cancer exosomal L1CAM contributes to migration through similar mechanisms as their corresponding non-exosomal counterparts, through integrin interaction and subsequent FAK or ERK phosphorylation [16,40,41].

### 2.3. Intracellular Soluble L1CAM (L1CAM-CT)

Inhibition of L1CAM cleavage, and a cytoplasmic domain-deleted version of L1CAM abrogated in L1CAM-induced gene transcription suggest a specific role for cytoplasmic cleavage fragment in gene regulation [27]. Five studies directly addressed the function of the intracellular 28 kD cytosolic cleavage fragment that is released upon proteolytic cleavage (L1CAM-CT) [15,16,23,26,79].

It was reported that apoptotic vesicles, but not exosomal vesicles, contain soluble L1CAM-CT (28 kD) [16], but it is not clear whether this vesicle-associated L1CAM-CT can be taken up and functionally affect other cells. Expression of L1CAM-CT induces transcription of pro-invasive proteins such as β3 integrin and cathepsin B, and downregulation of the tumor suppressor CRAPBPII, in ovarian and pancreatic cancer cells [15]. ERK activation, however, did not depend on L1CAM cleavage in these cells [15], which is consistent with findings in colorectal and pancreas cancer, that showed that expression of L1CAM-CT is, in contrast to L1CAM-FL, not sufficient to stimulate proliferation or metastasis [23,26].

In glioblastoma, nuclear translocation of L1CAM-CT is required for radioresistance. This is mediated by L1CAM-CT-dependent transcription of Myc, and downstream expression of NBS1. NBS1 is required for activation of the checkpoint protein ATM kinase which protects stem-cell like tumor subset from radiation [79].

### 2.4. Algorithm for Clinical Use of L1CAM 

The studies included in this review are predominantly preclinical studies, in which different assays to assess L1CAM expression were employed, including Western Blot, immunohistochemistry, immunofluoresence, flow cytometry, qPCR and RT-PCR (Appendix A). Figure 4 provides an inventory of antibodies, and corresponding epitopes within the L1CAM domains that have been used in the included studies. We have used this information to categorize the studies with the aim to understand the domain-specific functions of L1CAM. For future analysis of both preclinical and clinical studies, we provide an algorithm as illustrated in Figure 5 as guideline to define the form of L1CAM being expressed, in order to improve the use of L1CAM as a predictive, prognostic or therapeutic target.

## 3. Discussion

To improve the knowledge of L1CAM as prognostic marker and therapeutic target we have reviewed the contribution of different domain-specific L1CAM forms in relation to cancer progression.

The results presented here demonstrate that L1CAM has an intrinsic capability to regulate cellular processes through its cytoplasmic domain. Main pathways induced by the cytoplasmic domain of L1CAM-FL are mediated through an interaction with the cytoskeleton linker protein ezrin, which induces downstream effectors such as ERK and nuclear activation of NF-κB, although this can also be ezrin-independent. Nuclear signaling also occurs upon cleavage, which allows cytosolic L1CAM-CT to translocate to the nucleus to induce pro-invasive and therapy-resistance gene transcription profiles. So far it is not clear if both pathways induce similar target gene transcription or pro-tumorigenic events, which is highly relevant for designing therapeutic strategies. Due to its intracellular location, L1CAM-(FL)-CT-induced processes likely correlate with L1CAM expression levels in the same cell, although it is still possible that soluble L1CAM-CT-mediated effects can be induced by uptake of exosomal or apoptotic vesicles derived from other cells type. The extent to which CT-induced processes rely on ectodomain engagement appears model-dependent, since in pancreatic adenocarcinoma L1CAM-FL-CT-induced ERK activation depends integrin binding [23] or can in the absence of a functional L1CAM ectodomain be induced by CT-domain clustering [72], whereas ERK activation is integrin-independent in HEK293 cells [69].

The mechanisms by which L1CAM ectodomains can drive cellular processes are much more varied and may include cis- and trans homophilic and heterophilic interactions, and short and long-range paracrine effects of circulating shed ectodomains or exosomal forms. Of these, the L1CAM-integrin interactions, as mediated by interaction of the RGD motif in the 6th Ig domain are most extensively studied and appear to play a dominant role in L1CAM signaling. L1CAM-integrin interactions can control interaction of tumor cells with other cell types, including endothelia, mesothelia, nerves, and cancer-associated fibroblasts, allowing for heterotypic cell adhesions that could potentially promote tissue invasion and intravasation into the vasculature. Given the widely reported interactions with integrins across the model systems covered in this review, it is not surprising that several pathways typically associated with integrin activation, such as activation of Src, FAK and PI3K/Akt are often demonstrated. It is therefore likely that many of these pathways activated by L1CAM ectodomains represent effects caused by L1CAM engagement of integrins, potentially in concert with L1CAM-ectodomain induced signaling. In addition, heterophilic interactions of the soluble L1CAM-ECD with the FGFR was demonstrated, leading to FGFR-dependent migration and proliferation. 

Shedding of L1CAM-ECD appears a common tumor-progression-associated mechanism. The studies covered in the review demonstrate abundant in vitro evidence for indirect tumor promoting effects of L1CAM ectodomains, and the challenge will be to understand its implications in patients. Specifically, it must be kept in mind that cell-associated L1CAM staining, as observed in IHC, or increased L1CAM transcription, as assayed by mRNA analyses, does not necessarily directly reflect L1CAM-induced tumor-promoting signaling in the same cell. 

Thus, the question how domain-specific forms of L1CAM contribute to cancer progression can only be partially answered, challenging L1CAM’s clinical applicability. A remaining challenge is how to translate the knowledge on LICAM domain and domain-specific functions from this review towards clinical practice. A recent systematic review and meta-analysis of 37 studies by Hua indicated L1CAM to be an important prognostic factor for a range of tumors. The studies included in this meta-analysis predominantly assessed L1CAM expression by IHC staining using ectodomain-reacting antibodies (UJ127.11 and L1-14.10), whereas a minority of studies used RT-PCR or ELISA for detection [20]. As demonstrated in our algorithm, IHC is not sufficient to specify the form of LICAM that is expressed. More specifically, IHC with ectodomain-specific antibodies may recognize: L1CAM-FL, L1CAM-ΔCT, and circulating N-terminal cleavage products or exosomal forms. Yet, IHC provides insight into the subcellular localization of L1CAM. On the other hand, ELISA generally relies on commercial kits with proprietary antibodies towards distinct but undisclosed domains and will detect circulating extracellular forms of L1CAM, including the cleaved ectodomain, and most exosomal forms. Finally, PCR analyses will not yield any information on protein processing, but only assesses transcriptional level of L1CAM-FL mRNA’s, and lack information on resulting protein levels and the extent of proteolytic processing. Thus, results from clinical analyses should be considered in the context of the tumor microenvironment and along with activation of other pathways, such as activation of integrins, growth factor receptors, and L1CAM-intrinsic pathways. This information may reveal soluble versions of L1CAM act locally of have also has systemic effect on other cell types by engaging in homo- or heterotypic interactions. In this perspective, the algorithm provided could contribute in standardization of future studies reporting on L1CAM.

The strength of this review includes the comprehensive evaluation of all available literature on L1CAM functionality and the structural categorization according to the domain and domain-specific molecular mechanism of L1CAM. It provides insights into the contributions of the L1CAM forms to the different stages of cancer progression, including cell proliferation, cell adhesion and migration, with specific information of cell types used in in vitro studies, the type of interference to understand L1CAM domain-specific functions, and specific antibodies that allow assessment of different L1CAM forms. Although the findings of this study are promising, some limitations need to be addressed. Most studies focused on L1CAM-FL and lack comparison with other forms of L1CAM impeding comparison of the specific contributions of these forms. As described, assays were heterogeneous throughout the different studies and confirmation of the domain-specific L1CAM form was not always sufficient to draw definite conclusions about the forms of L1CAM investigated. Furthermore, insufficient knowledge on mechanisms of L1CAM transcriptional regulation, alternative splicing, and cleavage may hamper a complete overview of the array of L1CAM forms expressed at certain experimental conditions but also in vivo. In particular the regulation of upstream pathways controlling expression of L1CAM, for instance in response to therapy or by the tumor microenvironment [26,44,50,58,59,68,72,80] has not been specifically addressed, but are a relevant and an emerging field and it will be important to understand if this transcriptional upregulation is also coupled to altered posttranscriptional processing and function, as for instance was shown for altered shedding [16,26,33,41], splicing [62] or cleavage due to tumor-associated changes in glycosylation [39].

Finally, due to the strict selection criteria we chose to apply, many L1CAM-mediated pathways that have been identified in non-cancer cells were left out of this systematic review, even though those will likely have direct relevance for cancer progression. Thus, recent interesting findings that L1CAM, through PI3 and ERK, can promote cell surface sialyation and fucosylation in CHO cells [81] and stem cells [82], despite the rapidly emerging relevance of the glycocalyx in tumor biology [83]. In addition, interactions of L1CAM with growth factor receptors leading to receptor activation, is highly relevant for understanding cancer progression downstream of these receptors and has been studied in a much greater detail in non-cancerous cells [84,85,86,87] as compared to tumor cells [74]. Also, the roles of L1CAM interaction with ephrins [88], and potential downstream migration guidance regulation in tumors [89] and the occurrence and metabolic function of transmembrane-containing cleavage forms in mitochondria [90] in tumor cells await further investigation.

## 4. Material and Methods

### 4.1. Search Strategy

A systematic review was conducted according to the Meta-Analyses and Systematic Reviews of Observational Studies (MOOSE) guidelines [91] (Figure 2). The literature search was performed in the databases PubMed, Medline and Embase. All known synonyms of L1CAM and cancer were used and an overview of the search strategy can be found in Appendix A.

### 4.2. Study Selection

In total, 971 studies published between 2001 and July 2019 were generated through the database searches and cross referencing. Searching through Google Scholar did not further supplement the database searches. After removal of duplicates, 697 articles remained. Out of these, 97 articles were regarded as eligible. Full-text selection resulted in an additional exclusion of 52 studies resulting in a total of 45 studies for analysis (Figure A1). The following inclusion criteria were specified: focus on either L1CAM-FL in the plasma membrane intracellular L1CAM or extracellular L1CAM; elaborations on downstream signaling pathways; focus on cancer progression. Exclusion criteria were: review articles, non-English articles, non-malignancy focus, and studies that did not specify signaling pathways involved. In case of disagreement, a third investigator (JP or MZ) was consulted for a final decision. Two independent investigators (MM, CR) selected eligible articles based upon title and abstract. After this first selection, all remaining articles underwent full-text screening and it was decided whether these studies fulfilled the inclusion criteria.

### 4.3. Data Analysis

Following the selection process, the studies were analyzed and categorized into different categories based on intra- or extracellular location and the presumed forms of L1CAM being researched. The locational categories were the plasma membrane, the intracellular environment or the extracellular environment. If the form of L1CAM was not mentioned specifically, structural cues, such as involvement of the arginyl-glycyl-aspartic acid (RGD) sequence, specific binding sites or used antibodies, were used to determine to which category the studies belonged (Appendix A). If studies discussed multiple locations or forms, the results were mentioned any of the appropriate categories. For each category, the major cellular signaling players and their contribution to cancer progression were discussed.

## 5. Conclusions

In conclusion, L1CAM has an important role in cancer progression and its main functions can be separated into biological processes and signaling initiated directly by the L1CAM molecule, and the activation of processes that rely on autocrine- and paracrine signaling (Figure 3). Direct, cell-intrinsic L1CAM-mediated signaling is mediated through the cytoplasmic domain of L1CAM-FL, which is mostly associated with activation of ERK, often through ezrin. ERK-mediated signaling can lead to proliferation, and gene regulation. Upon cleavage, the soluble cytoplasmic domain transfers to the nucleus and also promotes gene transcription through ERK-independent mechanisms. The L1CAM-ECD is often associated with activation of integrin-dependent processes, including integrin-dependent migration and cell survival, which can be either mediated by L1CAM-FL-ECD, or by its soluble ECD. In addition, the L1CAM-ECD can bind and activate FGFR1, which was mostly reported for soluble ECD version, and is it not clear if L1CAM-FL can activate FGFR1 to a similar extent. Most studies focus on the full-length plasma membrane L1CAM, yet for selective targeting treatment knowledge on the domain-specific forms is a prerequisite. The L1CAM assay algorithm facilitates interpretation and translation into clinical practice.

## Figures and Tables

**Figure 1 ijms-20-04180-f001:**
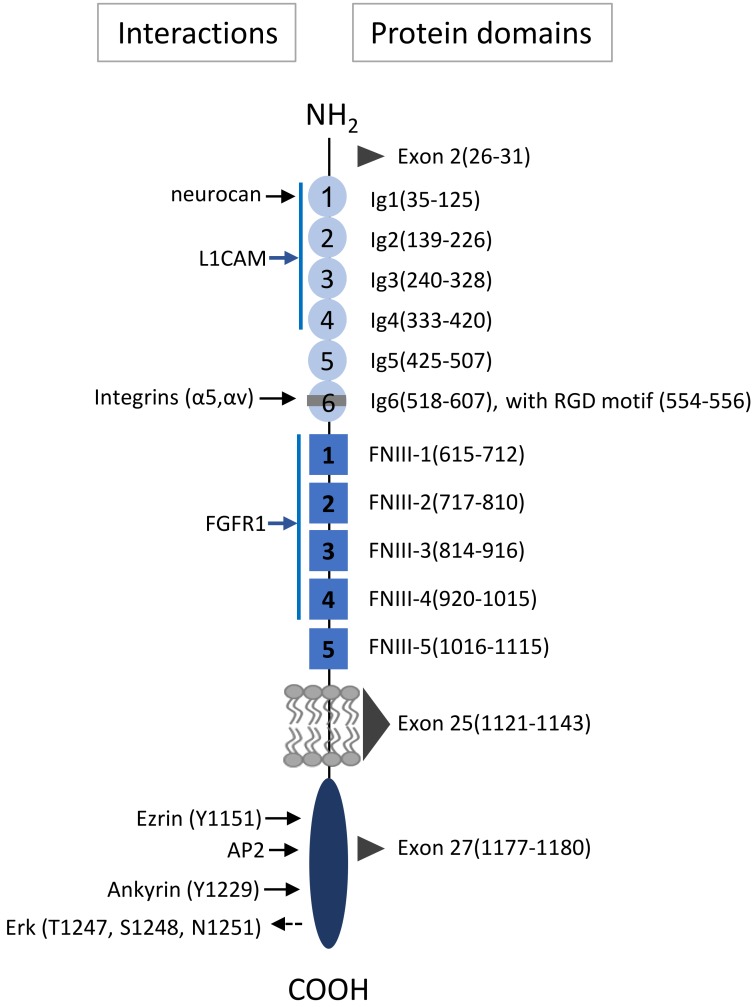
L1CAM domain structure with known interaction sites. On the left, sites of direct interaction are indicated with solid black right-pointing arrows, and possible indirect activation with dashed left-pointing arrow. Arrows in blue indicate interactions that span several Ig (immunoglobulin like)- or FNIII (fibronectin type III) repeats. On the right, numbered amino acid residues, corresponding with different domains or repeats are indicated. The black triangles indicate the amino acid sequences encoded by alternatively spliced exons 2, 25 and 27.

**Figure 2 ijms-20-04180-f002:**
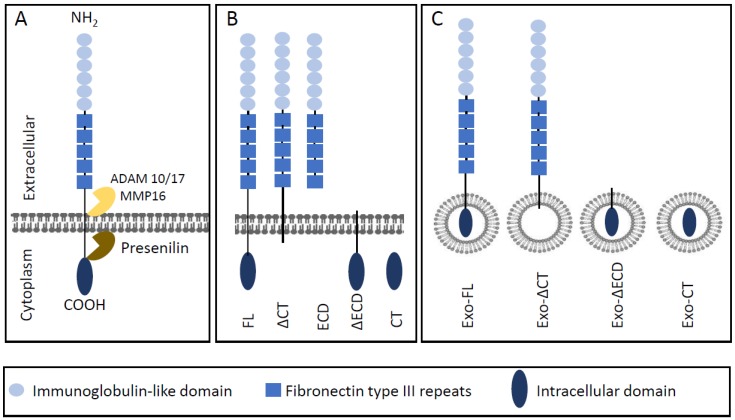
L1CAM cleavage and nomenclature of L1CAM forms discussed in this review. (**A**) Structure of L1CAM including main cleavage sites. ADAM: A Disintegrin and metalloproteinase domain-containing protein. MMP: matrix metalloprotease. (**B**) Full-length L1CAM (FL) and forms resulting from proteolytic cleavage. ΔCT lacks the C-terminal domain, NT represents the soluble N-terminal domain, ΔNT lacks the N-terminal domain, CT represents the cytosolic C-terminal cleavage product. (**C**) Shows exosomal forms (Exo) corresponding with the full-length or proteolytic cleavage forms indicated in B.

**Figure 3 ijms-20-04180-f003:**
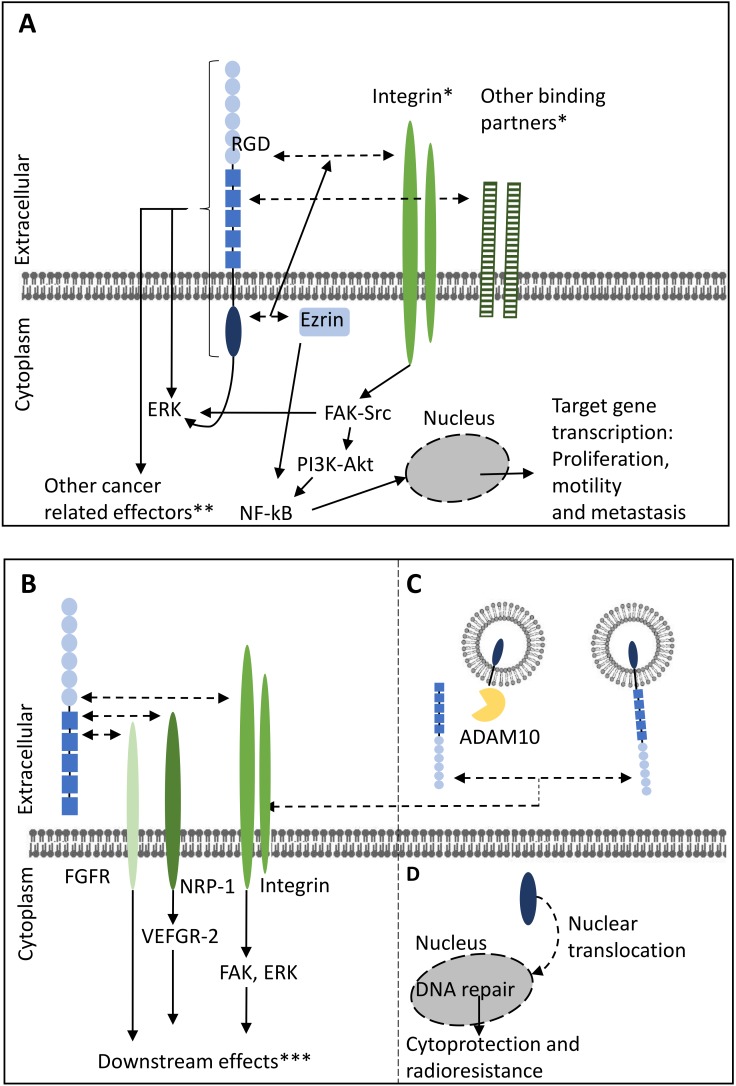
Schematic overview of the signaling pathways of full-length L1CAM in the plasma membrane and soluble forms of L1CAM and contribution to cancer progression-associated events. (**A**) Full-length L1CAM in the plasma membrane. (**B**) Soluble ectodomain of L1CAM. (**C**) Exosomal L1CAM. (**D**) Intracellular L1CAM. * Interactions between L1CAM and binding partners can occur in cis as well as in trans (not depicted). ** For example, JNK, Wnt-related effectors and caspase 8. *** Such as cell motility, proliferation and angiogenesis. Solid arrows indicate induction of effects. Dashed arrows indicate interactions possibilities. ADAM: A Disintegrin and metalloproteinase domain-containing protein, Akt: protein kinase B, ERK: extracellular signal-regulated kinase, FAK: focal adhesion kinase, FGFR: fibroblast growth factor receptor, JNK: c-jun n-terminal kinase, MMPs: matrixmetalloproteinases, NF-κB: nuclear kappa-light-chain-enhancer of activated B cells, NRP: neuropilin, PI3K: phosphatidylinositide 3-kinase, RGD: Arginylglycylaspartic acid, STAT: signal transducer and activator of transcription, VEGFR: vascular endothelial growth factor receptor, Wnt: wingless-related integration site.

**Figure 4 ijms-20-04180-f004:**
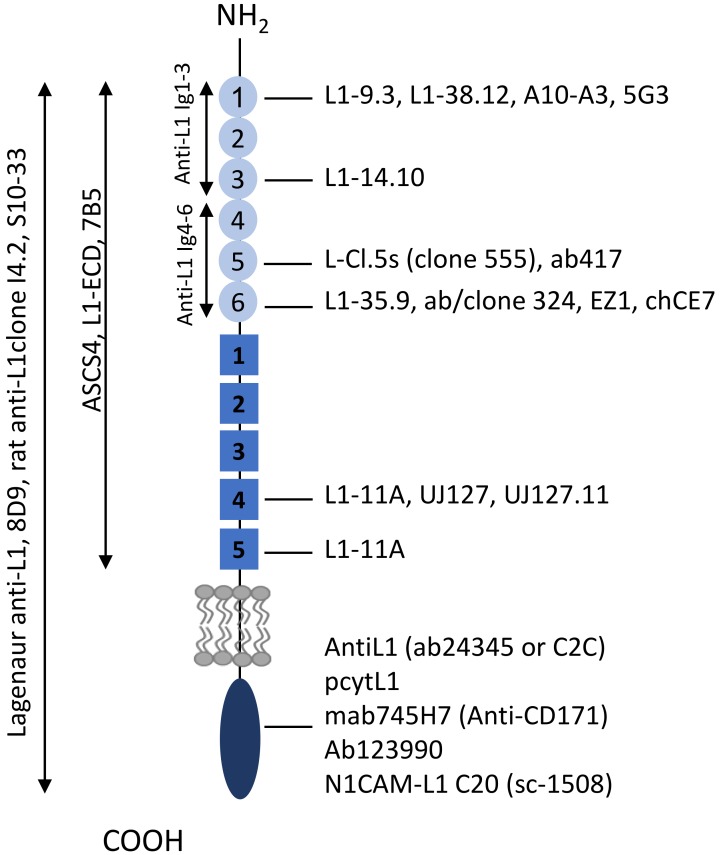
Overview of epitopes of anti-L1CAM antibodies. See Figure 1 for an overview of amino acid sequences corresponding with the indicated epitopes.

**Figure 5 ijms-20-04180-f005:**
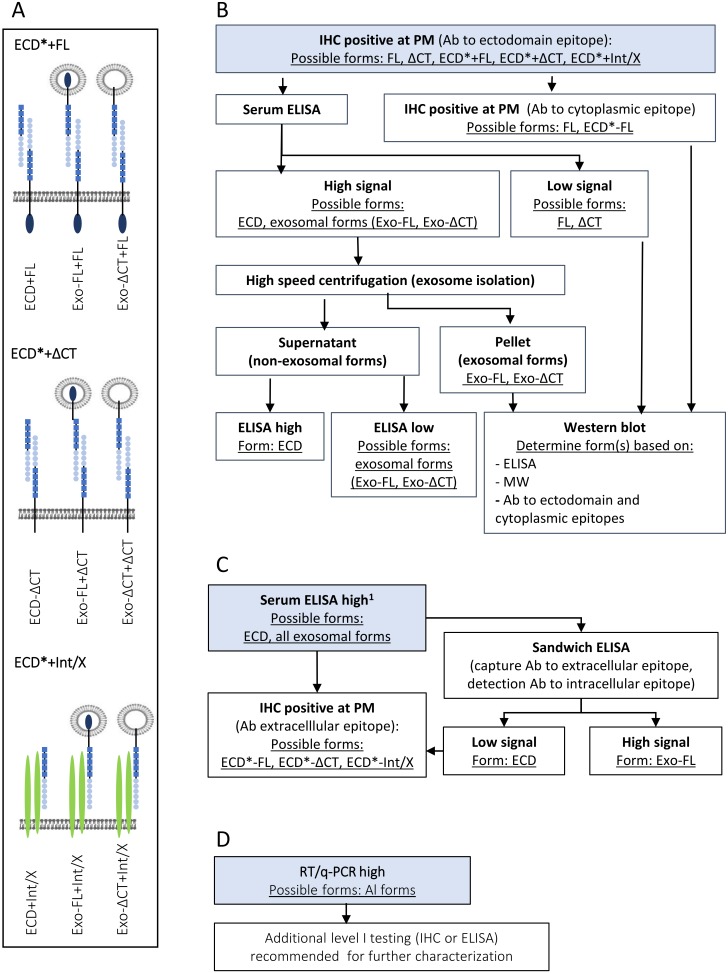
Suggested algorithm for the assessment of the form and location of L1CAM based on common clinical analyses of L1CAM expression. (**A**) Analysis of L1CAM is most commonly assessed by IHC using ECD-domain-reacting antibodies. In addition to recognizing L1CAM-FL or L1CAM-ΔCT, positive staining at the plasma membrane can represent homotypic interactions of L1CAM-FL or L1CAM-ΔCT with soluble ECD, or with exosomal forms expressing the ECD (collectively indicated ECD * + FL or ECD * + ΔCT). Positive staining at the plasma membrane may also be due to heterotypic interactions of ECD-containing soluble or exosomal L1CAM forms with other proteins, such as integrins (Int) or other undefined proteins (X). These heterotypic interactions are indicated with ECD * + Int/X. (**B**) Suggested diagnostic algorithm based on initial positive IHC staining with an ECD-domain-reacting antibody. Suggested follow-up analysis should includes ELISA analysis of serum to determine the level of soluble forms. The presence of FL can be assessed by IHC using a CT-domain-specific antibody, the staining of which should overlap with ECD-specific antibody staining in case of expression of FL. More detailed analyses to distinguish soluble forms can be done by high speed centrifugation of serum samples, by which soluble ECD and exosomal forms can be distinguished. Additional western blot and ELISA analysis using ECD- and CT-reactive antibodies can further stratify exosomal forms by analysis of molecular weight and the presence or absence of either domain. Ab is antibody. Nomenclature of L1CAM forms is also given in the legends of Figure 1C and Figure 2B. (**C**) Algorithm based on initial assessment of L1CAM in serum by ELISA. ELISA outcome depends on type of ELISA and on the antibodies used. This algorithm assumes direct ELISA, or a sandwich ELISA with either or both antibodies reacting with the ECD, or a commercial assay with proprietary information regarding antibodies. Downstream analyses to distinguish CT-containing forms from FL may include sandwich ELISA combining CT- and ECD-reacting antibodies, and/or IHC with a ECD-reacting antibody, which may be expanded to additional analyses as indicated in B. (**D**) Increased L1CAM expression as determined by RT/qPCR will not provide information on processing and cleavage form of L1CAM and additional analyses by IHC and ELISA is recommended.

**Table 1 ijms-20-04180-t001:** Summary of the main signaling pathways involved in domain-specific L1CAM signaling and the number of studies evaluated per category.

Form of L1CAM 45 Studies	Integrins	FAK/Src	PI3K/Akt	FGFR	Ezrin	ERK	NF-κB	Other *
**L1CAM in plasma membrane** **(L1CAM-FL)**								
Domain unspecified (*n* = 19)	√	√	√	√	√	√	√	√
Alternatively spliced variants (*n* = 3)				√				
Ectodomain (FL-CT) (*n* = 12)	√	√	√			√	√	√
Cytoplasmic domain (FL-ECD) (*n* = 9)					√	√	√	√
**Intracellular L1CAM**								
Cytoplasmic domain (CT) (*n* = 5)								√
**Extracellular L1CAM**								
Soluble ectodomain (ECD) (*n* = 12)	√	√		√				√
Exosomal L1CAM (Exo-FL, Exo-∆CT, Exo-∆ECD, Exo-CT) (*n* = 3)	√	√				√		

* Including activation-regulation of JNK, p38 MAPK, JAK/STAT, Wnt/β-catenin, caspases 3/7, MMP2 and MMP9 and transcriptional regulation of Myc, IL-1β, cathepsin B and L, MMP2 and MMP9, and caspase 8, MAGE, Wnt targets. See text for further explanation. Number in brackets indicate number of studies evaluated for each specific category. Some studies are represented in multiple categories. Akt: protein kinase B, ERK: extracellular signal-regulated kinase, FAK: focal adhesion kinase, FGFR; fibroblast growth factor receptor, NF-κB: nuclear kappa-light-chain-enhancer of activated B cells, PI3K: phosphatidylinositide 3-kinase.

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
