# Peer review of "L1 Cell Adhesion Molecule in Cancer, a Systematic Review on Domain-Specific Functions"

_ijms, 2019, doi:10.3390/ijms20174180_

Round 1

Reviewer 1 Report

In this review, Miriam van der Maten and colleagues summarise the roles that different forms of L1 (full length, soluble extracellular domain, cytosolic intracellular domain, L1 in exosomes) play in cancer. The review will be a valuable contribution to the field. However, I have several concerns that need to be addressed.  

My major concern is that the review lacks essential details. Descriptions of the interactions of L1 with different binding partners and signalling pathways activated by L1 should include essential details about types of cancer/tissue/cells where these interactions and signalling events were characterised. A discussion of how common they are for different cancers should be included.  

The review should include information about how different forms of L1 were identified and studied. In studies without a specific focus of L1 forms (unspecified L1CAM), how the functions of L1 were analysed? Were the antibodies against specific L1 domains used?  

Other points:

line 36: “neurogenetic” is an incorrect term and should be revised.

line 37: Reference 2 is focused on cancer. Appropriate references on the role of L1 in neurite outgrowth, fasciculation, cell adhesion, migration, survival, myelination, and synaptic plasticity should be given.  

Figure 1: ADAM10 and presinilin need to be described in the text and appropriate references must be given. A consistent style (as in panel A) should be used to depict the structure of L1 in all panels. The use of “NT” is misleading, because the whole extracellular domain is meant. Another abbreviation, eg “ED” should be used. What is the evidence that L1 in exosomes is positioned with its intracellular domain in the lumen? Appropriate references are needed to support this.  

line 61: “(1)” is this a reference?

line 67: “domain-specific L1CAM” – this phrase should be revised, eg “knowledge about specific domains of L1CAM”

line 88, 111, 137: numbers of “main findings” are given. Which criteria were used to decide which findings were “main findings” and can the authors be more specific and list them?  

Figure 3: integrins should be shown as dimers.  

In Appendix 2, consistent terminology should be used. What is the difference between “transfection” vs “overexpression” (by transfection?), “KD” vs “silencing” vs “downregulation (transfection)”, “WB” vs “immunoblot”. “Mutants”, “fusion proteins” should be explained.  

Author Response

Comments and Suggestions for Authors

In this review, Miriam van der Maten and colleagues summarise the roles that different forms of L1 (full length, soluble extracellular domain, cytosolic intracellular domain, L1 in exosomes) play in cancer. The review will be a valuable contribution to the field. However, I have several concerns that need to be addressed.  

My major concern is that the review lacks essential details. Descriptions of the interactions of L1 with different binding partners and signalling pathways activated by L1 should include essential details about types of cancer/tissue/cells where these interactions and signalling events were characterised. A discussion of how common they are for different cancers should be included.  

We want to thank the reviewer for the time invested in reviewing the manuscript and providing the useful comments. We feel that the comments of the reviewer have significantly improved the manuscript. As a general note, we would like to mention that we have thoroughly revised the results and discussion section within the manuscript, with more in-depth focus on the involved signaling pathways, in relation to the types of cancer. The focus on different types of cancer involves the entire results section and all forms of L1CAM. We have expanded Appendix 2 and did not only specified all investigated cancer types, but also described all models/patient materials that were used, including specification of all cancer cell lines, as well as used in vivo materials.

The review should include information about how different forms of L1 were identified and studied. In studies without a specific focus of L1 forms (unspecified L1CAM), how the functions of L1 were analysed? Were the antibodies against specific L1 domains used?  

All techniques used to identify different forms of L1CAM are described in detail in the updated version of Appendix 2. In addition, we have briefly touched the techniques in the manuscript text, e.g.

Twelve studies addressed the function of the ECD of L1CAM-FL (FL-ECD), often with approaches that involved targeted mutations within this domain, flanked by KD experiments.” (line 152 – 153)

All antibodies are described in Appendix 2, and the specific L1CAM domains targeted by the antibodies can be deduced from the updated version of Figure 1 and 4, in which we now provide the L1CAM domain with known interactors (Figure 1) and, in parallel, known antibody recognition sites (Figure 4).

Other points:

line 36: “neurogenetic” is an incorrect term and should be revised.

We have adjusted this in the manuscript.

line 37: Reference 2 is focused on cancer. Appropriate references on the role of L1 in neurite outgrowth, fasciculation, cell adhesion, migration, survival, myelination, and synaptic plasticity should be given.

We have inserted appropriate references in the abovementioned text which relate to the role of L1CAM in neural development (line 36 – 41).

Figure 1: ADAM10 and presenilin need to be described in the text and appropriate references must be given.

We have described the proteolytic cleavage in the manuscript (line 51 – 56) by ADAM10, ADAM17 and MMP16 which produce the soluble ectodomain, and by presenilin which produces the soluble cytoplasmic domain. Moreover, we have inserted the corresponding references into this section. We have updated Figure 2 with ADAM17 and MMP16.

A consistent style (as in panel A) should be used to depict the structure of L1 in all panels.

We have adjusted this in the manuscript.

The use of “NT” is misleading, because the whole extracellular domain is meant. Another abbreviation, eg “ED” should be used.

We have adjusted this in the manuscript and used the term ‘ECD’ (extracellular domain).

What is the evidence that L1 in exosomes is positioned with its intracellular domain in the lumen? Appropriate references are needed to support this.  

We have adjusted this into the manuscript (lines 58 – 61). Both exosomes, derived from exocytosed multivesicular endosomes, and microvesicles, which bud directly from the plasma membrane, maintain an exoplasmic membrane topology, and exosomal L1CAM thus will still expose its ectodomain towards the extracellular environment, which is similar to cell-associated L1CAM. For this we refer to a study by Raposo et al.

line 61: “(1)” is this a reference?

We have removed this reference from the manuscript.

line 67: “domain-specific L1CAM” – this phrase should be revised, eg “knowledge about specific domains of L1CAM”

We have adjusted this in the manuscript.

line 88, 111, 137: numbers of “main findings” are given. Which criteria were used to decide which findings were “main findings” and can the authors be more specific and list them?  

Main findings were mentioned as such when they were found in at least two studies. However, because we feel this might be too narrow, we have expanded the manuscript with other findings and  rephrased this in  the new version of the manuscript.

Figure 3: integrins should be shown as dimers.  

We have adjusted this in Figure 3.

In Appendix 2, consistent terminology should be used. What is the difference between “transfection” vs “overexpression” (by transfection?), “KD” vs “silencing” vs “downregulation (transfection)”, “WB” vs “immunoblot”. “Mutants”, “fusion proteins” should be explained.  

We have adjusted Appendix 2 and used uniform terminology.

Reviewer 2 Report

The role of L1CAM in cancer is certainly a relevant and timely topic for a review article. In fact, L1CAM is emerging as a major driver in several steps of tumor development. The diverse functions of the molecule and of its different domains in the context of cell pathophysiology and signaling make L1CAM even a more intriguing protein to investigate. Based on these very premises, a review on the role of L1CAM in cancer should provide a picture as comprehensive as possible on the impact of the molecule and of its domains on tumor progression.
Unfortunately, this is not the case for the manuscript in its current version. While the abstract alludes to a "systematic" analysis of the literature and of the available knowledge, the article is a collection of superficial pieces of information and, in most cases overlooks relevant aspects of L1CAM biology and function. Each section of the review should be expanded and provide a more in-depth and critical description of the functional role of the various domains.
Just to exemplify this criticism (which however extends to virtually all paragraphs), the description of intracellular L1CAM is disappointingly poor and defective (a 5-line paragraph seems really reductive). By the way, it is not accurate to state that only one study investigated the role of the cytoplasmic domain. For example, Gast et al (Oncogene 2008) performed quite an insightful research in ovarian cancer.
As a side note, applicable to all sections of the manuscript, a lot of findings on domain-specific functions were obtained in non-cancer systems (e.g. neural or vascular cells), yet they have implications for tumor biology. The authors may want to incorporate at least some of these studies and address their relevance for cancer development.
Another point that should be properly discussed is the existence of various domains in the L1CAM molecule. The division proposed by the authors in intracellular and extracellular L1CAM is really too simplistic. The extracellular region of L1 is composed of Ig and FNIII-like domains and specific properties and activities (which are relevant to cancer biology) have been assigned to some of these modules, including some mentioned by the authors such as the binding of the FNIII modules I-V to FGFR1 (Kulahin et al, Mol Cell Neurosci 2008) or that of Ig6 to integrins (many publications). Even the transmembrane domain has been recently described as a substrate for alternative splicing resulting in the release of a novel, soluble isoform which could have implications in cancer progression (Angiolini et al, eLife 2019). A review on domain-specific functions of L1CAM should include this type of information.
Finally, it is quite unusual to see a review article divided into Introduction, Materials and Methods, Results and Discussion. If this not explicitly requested by IJMS, I would encourage a more appropriate structure, for example Introduction, all the various sections, Conclusion and/or Future Perspectives.

Download Manuscript

Download Manuscript            

Author Response

Comments and Suggestions for Authors

The role of L1CAM in cancer is certainly a relevant and timely topic for a review article. In fact, L1CAM is emerging as a major driver in several steps of tumor development. The diverse functions of the molecule and of its different domains in the context of cell pathophysiology and signaling make L1CAM even a more intriguing protein to investigate. Based on these very premises, a review on the role of L1CAM in cancer should provide a picture as comprehensive as possible on the impact of the molecule and of its domains on tumor progression.
Unfortunately, this is not the case for the manuscript in its current version. While the abstract alludes to a "systematic" analysis of the literature and of the available knowledge, the article is a collection of superficial pieces of information and, in most cases overlooks relevant aspects of L1CAM biology and function. Each section of the review should be expanded and provide a more in-depth and critical description of the functional role of the various domains. Just to exemplify this criticism (which however extends to virtually all paragraphs), the description of intracellular L1CAM is disappointingly poor and defective (a 5-line paragraph seems really reductive).

We want to thank the reviewer for the time invested in reviewing the manuscript and providing the useful comments. We feel that the comments of the reviewer have significantly improved the manuscript. As a general note, we would like to mention that we have thoroughly revised and expanded the results and discussion section within the manuscript, with more in-depth focus on the involved signaling pathways, in relation to the types of cancer. The focus on different types of cancer involves the entire results section and all forms of L1CAM. We have included seven additional studies, including those mentioned below by the reviewer. The section on soluble L1CAM has been expanded and four additional studies were identified. Also, the section on intracellular L1CAM has been expanded thoroughly and includes nine studies which specifically focused on the intracellular domain of L1CAM-FL.

The figures have been adjusted and describe in more detail the functional role of the various domains of L1CAM, including the amino acids as known binding sites for interactions, and the sites for alternative splicing. Furthermore, we have expanded appendix 2 with information on experimental approaches, which include the use of L1CAM truncation and point mutants, thus providing specific information of domain-specific functions.

By the way, it is not accurate to state that only one study investigated the role of the cytoplasmic domain. For example, Gast et al (Oncogene 2008) performed quite an insightful research in ovarian cancer.

As indicated above, we have significantly expanded the result section, but also would like to point out that we felt it important to distinguish the soluble cytoplasmic domain, which is produced after proteolytic cleavage, from the cytoplasmic domain within the full-length L1CAM (L1CAM-FL-CT), which is described in detail in section 2.1.3. In this section we describe nine studies that have investigated L1CAM-FL-CT, including Gast et al. (Oncogene 2008). We have included five studies that elaborate on the role of the soluble cytoplasmic domain (CT) in section 2.4 of the manuscript.

As a side note, applicable to all sections of the manuscript, a lot of findings on domain-specific functions were obtained in non-cancer systems (e.g. neural or vascular cells), yet they have implications for tumor biology. The authors may want to incorporate at least some of these studies and address their relevance for cancer development.

We agree that many biological functions of L1CAM in neuronal systems and in the vasculature are highly relevant. However, for this systematic review we have applied strict criteria for the study inclusion which were predefined according to the MOOSE guidelines for preparing a systematic review which are outlined in reference 91, by Stroup et al.  Amongst these criteria were the focus on cancer biology and exploration of downstream pathways linked to L1CAM. We mention non-cancer studies extensively in the introduction section and discussion section, but have not included them as part of the results.

We have specifically addressed this issue in the discussion section on page 8, lines 355-365, where we briefly mention some important effects of L1CAM in non-cancer systems to raise awareness for these important aspects as follows:

Finally, due to the strict selection criteria we chose to apply, many L1CAM-mediated pathways that have been identified in non-cancer cells were left out of this systematic review, even though those will likely have direct relevance for cancer progression. Thus, recent interesting findings that L1CAM, through PI3 and ERK, can promote cell surface sialyation and fucosylation in CHO cells  and stem cells, despite the rapidly emerging relevance of the glycocalyx in tumor biology. In addition, interactions of L1CAM with growth factor receptors leading to receptor activation, is highly relevant for understanding cancer progression downstream of these receptors, and has been studied in a much greater detail in non-cancerous cells as compared to tumor cells. Also, the roles of L1CAM interaction with ephrins, and potential downstream migration guidance regulation in tumors and the occurrence and metabolic function of transmembrane-containing cleavage forms in mitochondria in tumor cells await further investigation.”

Another point that should be properly discussed is the existence of various domains in the L1CAM molecule. The division proposed by the authors in intracellular and extracellular L1CAM is really too simplistic. The extracellular region of L1 is composed of Ig and FNIII-like domains and specific properties and activities (which are relevant to cancer biology) have been assigned to some of these modules, including some mentioned by the authors such as the binding of the FNIII modules I-V to FGFR1 (Kulahin et al, Mol Cell Neurosci 2008) or that of Ig6 to integrins (many publications).

We have now included a new figure (Figure 1), in which we provide the domain structure of L1CAM1 with interaction sites of main interactors, including FGFR1, the integrin-interacting RGD motif and homotypic interaction sites, as well as cytoplasmic residues important for interactors for downstream signaling, such as ezrin and ERK. Furthermore, we have indicated the sites affected by alternative splicing. Along with the expanded information in the appendix, which provide detailed information on sites of mutations, the revised manuscript now addresses domain-specific functions in much greater detail.

Even the transmembrane domain has been recently described as a substrate for alternative splicing resulting in the release of a novel, soluble isoform which could have implications in cancer progression (Angiolini et al, eLife 2019).

We have included three studies on two alternative splicing variants (lacking the exon 2- and 27-, and exon 25-encoded domain), which also includes the study mentioned above (lines 140 – 148):

“Three publications address the function of two isoforms of L1CAM that are generated due to alternative splicing of the L1CAM mRNA. It is reported that L1CAM-SV, which lacks exon 2 and 27, is co-expressed with L1CAM-FL, with L1CAM-SV being the dominant form in many tumor type. Both L1CAM-SV and L1CAM-FL can promote motility in breast cancer cells. Surprisingly however, L1CAM-FL, but not L1CAM-SV, confers metastasis in  ovarian cancer, CRC, fibrosarcoma, and T-cell lymphoma. A second, newly identified soluble isoform lacking the exon 25-encoded transmembrane domain, was recently identified in endothelial cells. This isoform is highly expressed in ovarian-cancer associated vessels and has an increased potential to induce angiogenesis, which relies on FGFR1 signaling.”

A review on domain-specific functions of L1CAM should include this type of information.
Finally, it is quite unusual to see a review article divided into Introduction, Materials and Methods, Results and Discussion. If this not explicitly requested by IJMS, I would encourage a more appropriate structure, for example Introduction, all the various sections, Conclusion and/or Future Perspectives.

We understand the confusion this structure may create for the reviewer, as it is indeed not very commonly used for reviews in the field of molecular and cellular biology. We have performed this systematic review according to the Meta-Analyses and Systematic Reviews of Observational Studies (MOOSE) guidelines, which were developed in order to improve to usefulness of systematic reviews, published in JAMA 2000 (A). The guidelines ought to improve transparency within the field of research by providing a checklist on items to report, including a detailed elaboration on the methods used. In accordance with these guidelines we have structured our manuscript with an Introduction, Material and Methods, Results and Discussion.

Stroup DF, Berlin JA, Morton SC, Olkin I, Williamson GD, Rennie D, Moher D, Becker BJ, Sipe TA, Thacker SB. Meta-analysis of observational studies in epidemiology: a proposal for reporting. Meta-analysis Of Observational Studies in Epidemiology (MOOSE) group. JAMA. 2000 Apr 19;283(15):2008-12.

Reviewer 3 Report

Comments concerning database searching and Fig.2:

1)     Due to the fact that in the Results (lines 70-75) the Authors describe the rule of article selection and the fact that Fig. 2 adds nothing more,  this figure should be moved to supplements.

2)     MEDLINE is the largest subset of PubMed. Why did the Authors define MEDLINE database as other sources? The Authors should also use Scopus and ScienceDirect in order to obtain more successful results.

3)     Time criterion for article search should be updated and include also the year 2019. Limitation to the time interval between 2001 and Sep 2018 does not include the latest findings.

Comments concerning subsections 2.1 and 2.3:

L1CAM is a heavily glycosylated protein and its membrane expression depends on its ectodomain glycosylation. Neoplastic transformation is inextricably linked with the changes in glycosylation. It is well known that alteration in L1CAM glycosylation pattern influences L1CAM proteolytic cleavage, cell migration and invasion, and could help to distinguish between primary and metastatic stage of cancer. While describing the function of the extracellular domain, the Authors should also take into account the effect of L1CAM glycosylation on the behavior of tumor cells. The introduction of information on posttranslational modyfications of  L1CAM would enrich this manuscript with new information compared to the recently published reviews.

Comments concerning Fig.4:

The Authors should indicate the amino acids recognized by a particular antibody (where it is known), because, e.g., mAb74-5H7 only binds to L1CAM when Y1176 is dephosphorylated. Moreover, this antibody binds to L1CAM in clathrin coated pits and vesicles.

Minor comment:

1.      There are some printing errors in the text.

2.      Some text areas are marked in yellow or underlined.

Author Response

Comments and Suggestions for Authors

Comments concerning database searching and Fig.2:

1)     Due to the fact that in the Results (lines 70-75) the Authors describe the rule of article selection and the fact that Fig. 2 adds nothing more,  this figure should be moved to supplements.

We want to thank the reviewer for the time invested in reviewing the manuscript and providing the useful comments. We feel that the comments of the reviewer have significantly improved the manuscript. We have moved this figure to the supplements (supplementary figure 1).

2)     MEDLINE is the largest subset of PubMed. Why did the Authors define MEDLINE database as other sources? The Authors should also use Scopus and ScienceDirect in order to obtain more successful results.

The literature search was performed in Pubmed and Embase, and was designed according to the MOOSE guidelines and in close collaboration with an academic librarian with extensive experience in performing systematic reviews, published in JAMA 2000 (A). We therefore believe our search strategy has been adequate.  Prompted by the reviewer comments, we have performed an second screen of the publications that were derived by our search, and have now included several publications that were initially rejected.

Stroup DF, Berlin JA, Morton SC, Olkin I, Williamson GD, Rennie D, Moher D, Becker BJ, Sipe TA, Thacker SB. Meta-analysis of observational studies in epidemiology: a proposal for reporting. Meta-analysis Of Observational Studies in Epidemiology (MOOSE) group. JAMA. 2000 Apr 19;283(15):2008-12.

3)     Time criterion for article search should be updated and include also the year 2019. Limitation to the time interval between 2001 and Sep 2018 does not include the latest findings.

We have updated our literature search until July 2019.

Comments concerning subsections 2.1 and 2.3:

L1CAM is a heavily glycosylated protein and its membrane expression depends on its ectodomain glycosylation. Neoplastic transformation is inextricably linked with the changes in glycosylation. It is well known that alteration in L1CAM glycosylation pattern influences L1CAM proteolytic cleavage, cell migration and invasion, and could help to distinguish between primary and metastatic stage of cancer. While describing the function of the extracellular domain, the Authors should also take into account the effect of L1CAM glycosylation on the behavior of tumor cells.

The introduction of information on posttranslational modifications of  L1CAM would enrich this manuscript with new information compared to the recently published reviews.

Due to our inclusion criteria requiring information on both  tumor biology and at least some information on downstream cancer-associated signaling of L1CAM domains, we have initially not included reports on mechanism that control expression or processing of L1CAM. Since, as far as we could tell, no reports have been published that link L1CAM glycosylation directly to signaling in tumor cells, such studies were not included.  However, we agree this is a relevant topic, and have addressed this limitation in the discussion, where we briefly discuss relevant work that was not included in our selection. This includes a discussion on the role of glycosylation and other posttranslational modifications on p8, line 248-353 as follows:

 “In particular the regulation of upstream pathways controlling expression of L1CAM, for instance in response to therapy or by the tumor microenvironment [26,44,50,58,59,68,72,80] has not been specifically addressed, but are a relevant and an emerging field and it will be important to understand if this transcriptional upregulation is also coupled to altered posttranscriptional processing and function, as for instance was shown for altered shedding [16,26,33,41], splicing [62] or cleavage due to tumor-associated changes in glycosylation [39]”.

Comments concerning Fig.4:

The Authors should indicate the amino acids recognized by a particular antibody (where it is known), because, e.g., mAb74-5H7 only binds to L1CAM when Y1176 is dephosphorylated. Moreover, this antibody binds to L1CAM in clathrin coated pits and vesicles.

We have included a new figure (Figure 1), in which we provide the domain structure of L1CAM1 with interaction sites of main interactors, including FGFR1, the integrin-interacting RGD motif and homotypic interaction sites, as well as cytoplasmic residues important for interactors for downstream signaling , such as ezrin and ERK. Furthermore, we have indicated the sites affected by alternative splicing. This figure is flanked by a revised Figure 4, which indicates the domains recognized by specific antibodies. Cross-referencing both figures, along with the expanded information on antibodies used in the different studies, provides detailed information on the relevant antibodies.

Minor comment:

There are some printing errors in the text.

We have thoroughly screened the manuscript for printing errors and adjusted them when found.

Some text areas are marked in yellow or underlined.

We have adjusted this in the manuscript.

Reviewer 4 Report

This review article by van der Maten et al. focuses on the domain-specific functions of L1CAM in relation to cancer progression, and outlined pre-clinical assays used to assess L1CAM.  This review distills down relevant literature on L1CAM to include only 38 studies that met all of their criteria.  It is very well written and will be a good addition to the literature on L1CAM, especially because it addresses domain-specific functions in relation to cancer.  Although this article is good at “cutting to the chase”, it may be a little too brief in certain sections.  This may or may not be due to exclusion of papers because of their strict criteria.  For example, section 2.2 on intracellular L1CAM is very brief, and excludes some papers that probably ought to be included.  Similarly, section 2.3.1 on soluble L1 ectodomain is very brief, especially considering that it is a form that appears to be very relevant to cancer progression.  These sections also exclude papers that seem like they should have been included.  Some examples are linked below.  If such papers were excluded due to the authors’ strict selection criteria, then their criteria should be slightly broadened, or else these and other relevant studies should still be discussed outside of their tabulated 38 studies.  The studies listed below are only examples, and there may be others that also should be mentioned, if not included in the main group.  Discussion of additional relevant papers that might give additional context to their review, even if excluded from their strict selection criteria, would be a good thing.  Lastly, the Conclusions section ought to recap some of the specific conclusions that were made in the other sections, which should follow their statement that “L1CAM has an important role in cancer progression that can be attributed to domain-specific forms…”

https://www.ncbi.nlm.nih.gov/pubmed/8640837

https://www.ncbi.nlm.nih.gov/pubmed/17145883

https://www.ncbi.nlm.nih.gov/pubmed/26883759

Author Response

Comments and Suggestions for Authors

This review article by van der Maten et al. focuses on the domain-specific functions of L1CAM in relation to cancer progression, and outlined pre-clinical assays used to assess L1CAM.  This review distills down relevant literature on L1CAM to include only 38 studies that met all of their criteria.  It is very well written and will be a good addition to the literature on L1CAM, especially because it addresses domain-specific functions in relation to cancer.  Although this article is good at “cutting to the chase”, it may be a little too brief in certain sections.  This may or may not be due to exclusion of papers because of their strict criteria. 
For example, section 2.2 on intracellular L1CAM is very brief, and excludes some papers that probably ought to be included.  Similarly, section 2.3.1 on soluble L1 ectodomain is very brief, especially considering that it is a form that appears to be very relevant to cancer progression.  These sections also exclude papers that seem like they should have been included.  Some examples are linked below.  If such papers were excluded due to the authors’ strict selection criteria, then their criteria should be slightly broadened, or else these and other relevant studies should still be discussed outside of their tabulated 38 studies. The studies listed below are only examples, and there may be others that also should be mentioned, if not included in the main group.  Discussion of additional relevant papers that might give additional context to their review, even if excluded from their strict selection criteria, would be a good thing. 

We want to thank the reviewer for the time invested in reviewing the manuscript and providing the useful comments. We feel that the comments of the reviewer have significantly improved the manuscript.  As a general note, we would like to mention that we have thoroughly revised the results and discussion section within the manuscript, with more in-depth focus on the involved signaling pathways, in relation to the types of cancer. The focus on different types of cancer involves the entire results section and all forms of L1CAM. We have included seven additional studies, including those mentioned below by the reviewer. The section on soluble L1CAM has been expanded and four additional studies were identified. Also, the section on intracellular L1CAM has been expanded thoroughly and includes nine studies which specifically focused on the intracellular domain of L1CAM-FL. It however needs to be pointed out that we have applied strict criteria for the study inclusion which were predefined according to the MOOSE guidelines (A). Amongst these criteria were the focus on cancer biology (non-cancer studies are mentioned extensively in the introduction section, but are not included as part of the results), and exploration of downstream pathways linked to L1CAM.
We have specifically addressed this issue in the discussion section on page 8, line 345-365, and briefly mention some important regulatory mechanisms upstream and downstream of L1CAM in non-cancer systems to raise awareness for these important aspects and to highlight specific areas relevant for L1CAM functions in cancer as follows:

Furthermore, insufficient knowledge on mechanisms of L1CAM transcriptional regulation, alternative splicing, and cleavage may hamper a complete overview of the array of L1CAM forms expressed at certain experimental conditions but also in vivo. In particular the regulation of upstream pathways controlling expression of L1CAM, for instance in response to therapy or by the tumor microenvironment [26,44,50,58,59,68,72,80] has not been specifically addressed, but are a relevant and an emerging field and it will be important to understand if this transcriptional upregulation is also coupled to altered posttranscriptional processing and function, as for instance was shown for altered shedding [16,26,33,41], splicing [62] or cleavage due to tumor-associated changes in glycosylation [39].

Finally, due to the strict selection criteria we chose to apply, many L1CAM-mediated pathways that have been identified in non-cancer cells were left out of this systematic review, even though those will likely have direct relevance for cancer progression. Thus, recent interesting findings that L1CAM, through PI3 and ERK, can promote cell surface sialyation and fucosylation in CHO cells [81] and stem cells [82], despite the rapidly emerging relevance of the glycocalyx in tumor biology [83]. In addition, interactions of L1CAM with growth factor receptors leading to receptor activation, is highly relevant for understanding cancer progression downstream of these receptors, and has been studied in a much greater detail in non-cancerous cells [84-87] as compared to tumor cells [74]. Also, the roles of L1CAM interaction with ephrins [88], and potential downstream migration guidance regulation in tumors [89] and the occurrence and metabolic function of transmembrane-containing cleavage forms in mitochondria [90] in tumor cells await further investigation.”

The references suggested by the reviewer have now been included in this review, either as part of the general introduction (Izumoto, 1996), or have additionally been included in the final selection (Shtutman, 2007; Anderson, 2016)

https://www.ncbi.nlm.nih.gov/pubmed/8640837

https://www.ncbi.nlm.nih.gov/pubmed/17145883

 https://www.ncbi.nlm.nih.gov/pubmed/26883759

Lastly, the Conclusions section ought to recap some of the specific conclusions that were made in the other sections, which should follow their statement that “L1CAM has an important role in cancer progression that can be attributed to domain-specific forms…”

We have adjusted the conclusion section with the requested changes, where we recap the main findings. These main findings are also summarized in table 1.

Stroup DF, Berlin JA, Morton SC, Olkin I, Williamson GD, Rennie D, Moher D, Becker BJ, Sipe TA, Thacker SB. Meta-analysis of observational studies in epidemiology: a proposal for reporting. Meta-analysis Of Observational Studies in Epidemiology (MOOSE) group. JAMA. 2000 Apr 19;283(15):2008-12.

Round 2

Reviewer 2 Report

The Authors have addressed my concerns appropriately and have significantly improved the review, adding several features of L1CAM molecule domains and incorporating the relevant literature.